# Molecular-Genetic Biomarkers of Diabetic Macular Edema

**DOI:** 10.3390/jcm13237426

**Published:** 2024-12-05

**Authors:** Irene Andrés-Blasco, Alex Gallego-Martínez, Ricardo Pedro Casaroli-Marano, Salvatore Di Lauro, Jose Fernando Arévalo, Maria Dolores Pinazo-Durán

**Affiliations:** 1Ophthalmic Research Unit “Santiago Grisolía”/Fisabio, 46017 Valencia, Spain; irene.andres@fisabio.es (I.A.-B.); alex.gallego@fisabio.es (A.G.-M.); 2Cellular and Molecular Ophthalmo-Biology Group, Department of Surgery, Faculty of Medicina and Odontology, University of Valencia, 46017 Valencia, Spain; 3Research Network in Inflammatory Diseases and Immunopathology of Organs and Systems “REI-RICORS”, RD21/0002/0032, Institute of Health Carlos III, 28029 Madrid, Spain; arevalojf2020@gmail.com; 4Department of Surgery, School of Medicine and Hospital Clínic de Barcelona, Universitat de Barcelona, 08036 Barcelona, Spain; 5Department of Ophthalmology, University Clinic Hopital, 47003 Valladolid, Spain; sadilauro@live.it; 6Department of Ophthalmology, Wilmer Eye Institute, Johns Hopkins University, Baltimore, MD 21287, USA

**Keywords:** diabetic macular edema, biomarker, oxidative stress, apoptosis, inflammation, blood-retinal barrier

## Abstract

**Background:** Diabetic macular edema (DME) is a leading cause of vision impairment and blindness among diabetic patients, requiring effective diagnostic and monitoring strategies. This systematic review aims to synthesize current knowledge on molecular biomarkers associated with DME, focusing on their potential to improve diagnostic accuracy and disease management. **Methods:** A comprehensive search was conducted in PubMed, Embase, Medline, and the Cochrane Central Register of Controlled Trials, covering literature from 2004 to 2023. Out of 1074 articles initially identified, 48 relevant articles were included in this systematic review. **Results:** We found that molecules involved in several cellular processes, such as neuroinflammation, oxidative stress, vascular dysfunction, apoptosis, and cell-to-cell communication, exhibit differential expression profiles in various biological fluids when comparing diabetic individuals with or without macular edema. **Conclusions:** The study of these molecules could lead to the proper identification of specific biomarkers that may improve the diagnosis, prognosis, and therapeutic management of DME patients.

## 1. Introduction

Diabetes Mellitus (DM) has been established as a global pandemic, with more than 500 million adults suffering from this disease in 2021 according to the reports of International Diabetes Federation (IDF) Diabetes Atlas 10th edition (https://diabetesatlas.org, accessed on 16 February 2024). Additionally, a worrying growth in the diabetic population is being expected worldwide over the next years, as depicted in Table 1.

Amongst the several DM complications, diabetic retinopathy (DR) appears as one of the most prevalent, affecting up to one-third of DM patients and having been defined as a major global cause of visual impairment and blindness in working-age adults [1]. This pathological condition appears as a direct consequence of the hyperglycemic milieu that becomes established in diabetics [2], which disrupts different cell signaling processes, as depicted in Figure 1. These harmful stimuli lead to a dysfunction of the two main elements in the retina, the microvascular system and the nervous tissue, which ultimately causes the clinical signs observed in the ocular fundus of DR patients [3,4].

Vision loss due to DR may result from several mechanisms, but diabetic macular edema (DME) is undoubtedly the most common cause of visual impairment among DM patients, occurring in approximately 14% of diabetics and accounting for about three quarters of cases of visual loss [5]. From a mechanical perspective, DME appears as a result of an imbalance between fluid entry, fluid exit, and retinal hydraulic conductivity, which ultimately leads to the accumulation of intraretinal or subretinal fluid in the macular area, more precisely located in the inner nuclear layer (INL), outer plexiform layer (OPL), Henle’s fiber layer (HFL), and the subretinal space (SS) [6].

Under physiological conditions, Müller glia cells, the main structural and metabolic support cells of the retina, and retinal pigment epithelial (RPE) cells, located behind the retina, remove the fluid from the retinal interstitial tissue, keeping an optimal hydrostatic balance [7]. Additionally, the maintenance of the blood-retinal barrier (BRB) is responsible of regulating the fluid exchange between the vascular system and the retinal tissue [8]. However, DME alters these components as a result of harmful signaling processes that occur under the hyperglycemic conditions (Figure 1), leading to a series of chronographic cellular and tissular events that manifest themselves in the ocular fundus examination: (i) microaneurysms, (ii) dot and blot retinal hemorrhages, (iii) lipidic exudates (hard exudates: a result of chronic edema), and (iv) cotton wool spots (microinfarction zones of the retinal nerve fiber layer). In later stages, the ophthalmological exams show the presence of pivotal signs related to augmented fluid efflux, venous dilation, arterial narrowing, intraretinal microvascular abnormalities (IRMAs), macular edema, and chorio-retinal ischemia that can predispose preretinal vessel neovascularization located on the optic nerve, retinal surface and/or vitreous body. Altogether, these processes lead to progressive visual impairment [9,10,11,12,13,14,15].

Diabetics must be examined to screen for eye complications, carried out by methods including optical coherence tomography (OCT) retinal imaging, OCT angiography (OCTA), the autofluorescence of ocular fundus, stereoscopic fundus photographs, ultrawide field fundus imaging (UWFI), and fluorescein angiography (FA) [9,13,16,17,18,19,20,21,22,23]. Recently, the artificial-intelligence-based devices, such as the new machine learning OCT-omics prediction model, or the specific retinal software programs, have contributed to patient screening and the detection of different severities of DR [24,25,26].

As a multifactorial, complex, and disabling pathology, a variety of treatments are currently available. Pharmaceutical approaches consist of intravitreal therapy with intraocular injections of steroids and anti-angiogenic drugs [for inhibiting the vascular endothelial growth factor (VEGF) action], which can be complemented by other surgical care techniques, including laser photocoagulation and vitrectomy, sometimes with suboptimal results [4]. For this reason, early diagnosis is still the best approach for DME patients. However, the detection of the DR usually occurs once the retina is compromised, with the appearance of fundus signs. In this context, biomarkers, defined as biological parameters that allow the detection of the presence of a certain biological condition [11,27], may represent an interesting option for the management of DME patients. Biomarkers can potentially give us information about the disease even when we have not yet found clinical evidence of its presence.

In the present systematic review, we aimed to summarize the current understanding of which molecular biomarkers have been studied and validated in the context of DME, with the main objective of improving our battery of diagnostic and follow-up options and improving outcomes for diabetic patients.

## 2. Materials and Methods

### 2.1. PIO Analysis

According to the PIO (Participants/Intervention/Outcome) analysis guidelines, the present systematic review is based on the following research question: “are molecular biomarkers a good diagnostic tool for DME?”. The research question parts can be divided into 3 main components:Population. Patients with DME and type 2 diabetes mellitus (T2DM) from both sexes in any state of diabetic retinopathy and aged between 40 and 80 years old. Patients with DME that had type 1 diabetes mellitus (T1DM) and were aged older than 80 or younger than 40 years old were not the object of this review;Intervention. Molecular biomarkers involved in DME pathogenic pathways, including inflammatory, apoptotic, angiogenic, and neurodegenerative;Outcome. The outcome object of this review is the diagnosis of DME.

This systematic review was registered in the PROSPERO database (National Institute for Care and Health Research, NIHR) with the ID number CRD42023441129.

### 2.2. Literature Search and Processing Strategy

Using the PRISMA checklist (Figure 2), we conducted a systematic review of all studies published in peer-reviewed journals. We initially retrieved 1074 articles written in English from Medline (via PubMed), Embase, and the Cochrane Library with a date range from 2004 to 2023. We systematically searched the database by combining the following keywords: biomarker AND [caspase OR apoptosis OR oxidative stress OR neurodegeneration OR cytokine OR VEGF OR inflammation OR vascular dysfunction OR microRNA OR miRNA OR neurotrophic factor OR growth factor] AND [diabetic macular edema OR diabetic macular oedema]. Seventeen of the total initial publications were excluded due to duplication reasons. Of the 1057 identified studies, 958 were excluded for not following the inclusion criteria cited in Section 2.3. The second phase of eligibility resulted in the exclusion of 51 of the 99 pre-selected studies since they involved some type of treatment, did not include the population object of the systematic review, or involved exclusively non-molecular biomarkers. Following the PRISMA checklist resulted in the final selection of 48 studies that were included in the systematic review.

All records found with the search strategies were downloaded to a bibliographic manager (RefWorks^®^), then checked for duplicates and then screened according to the inclusion criteria by title and abstract. Two independent reviewers screened the records separately and compared their selection. If a discrepancy occurred, it was discussed in front of an expert. In the case of all studies in which there were any doubts as to whether they fulfilled the inclusion criteria, the article was fully recovered and reviewed in detail for selection criteria. During this process, all records that were duplicated, discarded, or selected were closely tracked.

### 2.3. Inclusion and Exclusion Criteria

The following inclusion and exclusion criteria were applied:By participants included: T2DM patients with DME from both sexes and diverse ethnicities in any state of DR, aged between 40 and 80 years. We excluded studies in which the population included patients with type 1 DM (T1DM) and DME, aged >80 years or <40 years. We also excluded studies that included any type of treatment outcome in their development;By type of study: we accepted research articles that were observational retrospective and/or prospective cross-sectional studies, observational retrospective and/or prospective longitudinal studies, and observational case/control studies. We excluded reviews and studies that included patients that did not fit the population inclusion criteria;By outcomes: we incorporated a study if it included biomarkers involved in the pathogenic pathways and/or clinical signs for DME development, including inflammatory, oxidative stress (OS), microRNAs (miRNAs), vascular dysfunction, apoptotic, neurodegeneration-related, and other types of molecules with a described role in the progression of the disease, as depicted in Figure 3.

### 2.4. Data Extraction

Data on the name of the first author, year, sample size, age of the participants, sample type, biomarker type (inflammation, OS, miRNAs, vascular dysfunction, apoptosis, trophic factors and others), biomarker name, and outcome of the biomarker were extracted for each included study. The authors A.G.-M. and I.A.-B. performed the data extraction simultaneously, sharing the found results and reaching a consensus with the study coordinators (J.F.A and M.D.P.-D.), as well as in the cases that were uncertain.

### 2.5. Quality Assessment

The forms include a checklist for quality based on a modified version of the Newcastle–Ottawa Scale [28]. This scoring system has three main criteria, each one of which includes different aspects:Selection.
(a)Representativeness of the sample;(b)Sample size;(c)Use of a validation measurement tool.
Study controls.
(a)The study controls the most important factor;(b)The study controls for any additional factor.
Exposure.
(a)Use of statistical tests.

## 3. Results

### 3.1. Description of Studies

A total of 48 studies were selected from 1074 screened titles that met the inclusion criteria described (Figure 2). Studies were divided into 6 case/control studies, 13 prospective studies, 14 retrospective studies, and 15 cross-sectional studies. The main outcomes extracted from each included study in this systematic review are depicted in Table 2, Table 3, Table 4 and Table 5 which display information related to the study itself.

### 3.2. Main Biomarkers in DME

The articles included in this study showed the importance of different serological, cellular-related and molecular biomarkers for the diagnosis and management of DME. These markers can be divided into several groups according to their nature (Figure 3).

#### 3.2.1. Vascular Dysfunction

Vascular alterations are a main trait of most diabetic complications, having an important role in the evolution of the disease. In the context of this systematic review, 20 out of the 48 included studies analyzed the expression of molecules related with vascular dysfunction in the context of DME. The most studied molecule was VEGF, with 13 studies showing significantly higher levels in aqueous humor, vitreous humor, or plasma [29,30,31,32,33,34,35,36,37,38,39,40,41] and one study reporting no difference in the aqueous humor expression of VEGF [42]. Related to this, higher expression levels of the VEGF Receptor 2 (VEGFR-2) in plasma samples of DME patients were also found [43].

Additionally, vascular adhesion molecules such as Intercellular Adhesion Molecule-1 (ICAM-1) and Vascular Cell Adhesion Molecule 1 (VCAM-1) were quantified in six different studies. More specifically, ICAM-1 expression levels were increased in vitreous [36,39] and aqueous humor [31,44,45] and plasma [46], whereas VCAM-1 was increased in the aqueous [31] and vitreous humor [36] samples of patients with DME.

Less common vascular-related molecules were described in two of the included studies. Jiang et al. studied the expression of hsa-miR-377-3p, a miRNA associated with vascular damage [47], reporting an increased expression in the plasma samples of DME patients. Finally, the study conducted by Kaya et al. found higher plasmatic expression levels of Chitinase-3-like-1 [CHI3L1] in patients with DME [48].

#### 3.2.2. Inflammation

Amongst the molecules involved in the inflammatory process related to DME, cytokines were the most prevalent group, being included in 19 of the 48 reviewed studies. Interleukin (IL)-1a2 [31], IL-6 [30,31,32,33,36,37,38,39,42,48,49,50], IL-8 [31,32,33,35,36,37,38,41,42], IL-27 [36], Tumor Necrosis Factor-alpha (TNF-α) [36,51], IFN-gamma-inducible Protein 10 (IP-10) [41,42,52], Monocyte Chemoattractant Protein-1 (MCP-1) [39,41,42,45,46], and monokine induced by gamma interferon (MIG) [31,41] showed increased levels in different biological fluids, including plasma and vitreous and aqueous humor samples. Oppositely, IL-10 [38,44] showed significantly lower levels in the aqueous humor and plasma samples of DME patients. Additionally, one study showed a significantly increased IL-1RA/IL-1β ratio in vitreous humor samples extracted from DME patients [52].

Cellular-associated biomarkers were also identified in 7 of the 48 reviewed articles, being linked to the DME condition. Zhu et al. identified a significantly higher percentage (%) of neutrophils, but a lower % of monocytes and lymphocytes in the peripheral blood of DME patients [53,54]. Additionally, Lee et al. [55] and Umazume et al. [41] found a higher expression of cluster of differentiation (CD) 14 in plasma, a specific marker of monocyte/macrophage activation. Dimitriou et al. found increased plasmatic values of Higher White Blood Cell Count (HWBCC) [56], whereas Gundogdu et al. found a higher Mean Platelet Volume (MPV) and Neutrophil to Lymphocyte Ratio (NLR) in plasma samples of DME patients [57]. These last authors also found an increased Systemic Immune Inflammation Index (SII), agreeing with the findings of Elbeyli et al. [58]. Lastly, Itoi et al. found a decreased Th1/Th2 ratio, a sign of the existence of a proinflammatory process in the plasma samples of DME patients [59].

Another group of inflammation-related biomarkers, identified in 13 of the 48 reviewed articles, include a set of soluble molecules that could participate in the development of DME. Zhang et al. found increased levels of angiopoietin-1 (ANG-1) and Tissue Inhibitors of Metalloproteinase-1 (TIMP-1) in DME plasma samples [43]. In relation to this, Xu et al. and Yin et al. described increased plasmatic levels of Angiopoietin-like 4 (ANGPLT4) in DME patients [60,61]. Kimura et al. described increased vitreous humor levels of ferritin, fibrinogen, procalcitonin, and Serum Amyloid P Component (SAP) in samples extracted from DME patients when compared to non-DME patients [62]. Jonas et al. described increased levels of metalloproteinase (MMP) 1 and 9, and Plasminogen Activator Inhibitor 1 (PAI-1) in aqueous humor samples of DME patients [31]. Finally, other studies showed increased plasmatic levels of C-Reactive Protein (CRP) [49,51], aqueous humor levels of erythropoietin (EPO) [63], increased aqueous humor levels of Glucose Regulator Protein 78 (GRP78) [29,64], increased plasmatic levels of homocysteine (Hcy) [46,65], and increased vitreous humor levels of lipocalin 2 [36].

Finally, 2 of the 48 reviewed articles described changes in lipid-related pro-inflammatory molecules. Jiang et al. and Rhee et al. described an increment of several lipid metabolites in aqueous humor samples and oxylipins in plasma samples of DME, respectively. More specifically, Jiang et al. described an alteration in the levels of linoleic acid, linolenic acid, sphingolipid, and glycerophospholipid [66], whereas Rhee showed a variation in the amount of plasmatic 12-oxoETE, 15-oxoETE, 9-oxoODE, and 20-carboxyleukotriene B4 in DME patients when compared with healthy subjects [67].

#### 3.2.3. Oxidative Stress

OS-related biomarkers were found in 3 out of the 48 studies included in the review. Our group described increased plasmatic and vitreous humor levels of malondialdehyde (MDA), 4-hidroxinonenal (4-HNE), and superoxide dismutase (SOD), related to decreased levels of catalase (CAT) in DME subjects [30]. Additionally, Kalayci et al. found increased plasma levels of Ischemia Modified Albumin (IMA) and Total Oxidant Status (TOS), and lower values of Total Antioxidant Status (TAS) [68], whereas Sabaner et al. also found higher values for TOS in plasma samples of DME patients [34].

#### 3.2.4. MicroRNAs

miRNAs have recently been established as a useful tool to diagnose certain pathological conditions, using different types of biological samples. In this systematic review, 3 out of the 48 included studies analyzed the expression of specific subsets of miRNAs and associated them with DME. Grieco et al. showed significantly lower levels of let-7c-5p, hsa-miR-200b-3p, hsa-miR-199a-3p, and hsa-miR-365-3p in plasma and aqueous humor samples [69], Jiang et al. described lower plasmatic expression of hsa-miR-377-3p [47], and Cho et al. reported lower aqueous humor expression of the miRNAs hsa-miR-185-5p, hsa-miR-17-5p, hsa-miR-20a-5p, hsa-miR-15b-5p, and hsa-miR-15a-5p [33].

#### 3.2.5. Trophic Factors

Molecules with a trophic function appeared in 7 out of the 48 included studies. Placental Growth Factor (PlGF) showed an increased concentration in aqueous humor obtained from DME patients, according to the results published by Cho et al., Kwon et al., and Jonas et al. [31,33,37]. Sabaner et al. and Jonas et al. showed higher expression of Fibroblast Growth Factor (FGF) in DME-derived plasma and aqueous humor samples, respectively [34,40]. The study conducted by Jonas et al. also showed higher aqueous humor expression levels of Epidermal Growth Factor (EGF), Human Growth Factor (HGF), and Transforming Growth Factor β (TGF-β) in DME patients when compared to the control group [31]. Lastly, Kim et al. showed an increase in the aqueous humor levels of Platelet-Derived Growth Factor AA (PDGF-AA) [42], whereas Zhang et al. reported a plasmatic rise of the PDGF-BB levels in samples obtained from DME patients [43].

#### 3.2.6. Apoptosis

Only 2 out of the 48 included studies in the review analyzed biomarkers related to the apoptotic phenomenon. Amongst these, Andrés-Blasco et al. described increased levels of caspase 3 (CAS-3) in vitreous and plasma samples of DME patients [30], whereas He et al. characterized decreased plasmatic and aqueous humor levels of the lncRNA SNHG5 in DME individuals when compared to the control group [70].

#### 3.2.7. Other

Lastly, 10 out of 48 articles included molecules that cannot be classified in the previous molecular groups. Dimitriou et al. described a decrease in the hematocrit and lipoprotein levels of DME-extracted plasma samples when compared to the control group [56]. The study conducted by Ji et al. indicates that DME patients have increased aqueous humor levels of Dickkopf 3 (DKK-3) [71]. Jiang et al. demonstrated the existence of higher concentrations of Clavulanic Acid (CA), alongside the decrease of several aminoacidic metabolites [setoclavine, atropine, d-synephrine, muscarine, I-dopachromate, α-methylphenylalanine, 2-(Formamido)-N1-(5-Phospho-d-ribosyl)] in aqueous humor samples of DME patients [66]. Similarly, Rhee et al. described lower levels of certain amino acids (asparagine, aspartic acid, glutamic acid, and lysine) in the plasma samples of DME patients [67]. The study of Kim et al. quantified the concentration levels of Vitamin D (VitD) in plasma and aqueous humor; in this sense, plasmatic VitD showed similar levels between DME patients and the control group, whereas aqueous humor VitD levels were higher in the DME group [72]. Plasmatic uric acid and citric acid levels were analyzed in two different studies with different outcomes: the study by Rhee et al. indicates that uric and citric acid levels are increased in plasma samples of DME patients [67], whereas Hu et al. described an increase in citric acid levels but a decrease in uric acid concentration [73]. Additionally, the study conducted by Naveen et al. seems to indicate that urea plasmatic levels are increased in DME patients when compared to healthy individuals [74]. Yenihayat et al. designed a study in which they proved that glycosylated hemoglobin (HbA1c) was increased in the vitreous humor samples of patients with DME [35]. Finally, Neuron-Specific Enolase (NSE) [75] and Antihexokynase 1 antibody (HK1-Ab) [76] were shown to have an increased expression in plasma samples obtained from DME patients.

**Table 2 jcm-13-07426-t002:** Characteristics of case-control studies (*n* = 6).

Reference	Sample Size (*n*)	Age, Ethnicity	Sample Type	Biomarker Type	Biomarker	Outcome
[30]	160	40–80, Spanish	P/VH	Vascular dysfunction	VEGF	↑
				Inflammation	IL-6	↑
				Oxidative stress	MDA	↑
					4-HNE	↑
					SOD	↓
					CAT	↑
				Apoptosis	CAS-3	↑
[49]	100	33–59, Indian	P	Inflammation	IL-6	↑
					CRP	=
[69]	30	57–79, Italian	P/AH	microRNAs	let-7c-5p	↓
					hsa-miR-200b-3p	↓
					hsa-miR-199a-3p	↓
					hsa-miR-365-3p	↓
[57]	120	55–59, Turkish	P	Inflammation	NV	↑
					MPV	↑
					NLR	↑
					SII	↑
[66]	60	40–80, Chinese	AH	Inflammation	Lipid metabolites	↓
				Other	CA	↑
					Aminoacid metabolites	↓
[31]	45	46–70, German	AH	Vascular dysfunction	ICAM-1	↑
					VCAM-1	↑
					VEGF	↑
				Inflammation	IL-1a2	↑
					IL-6	↑
					IL-8	↑
					IP-10	↑
					MCP-1	↑
					MIG	↑
					MMP-1	↑
					MMP-9	↑
					PAI-1	↑
				Trophic factors	EGF	↑
					HGF	↑
					PlGF	↑
					TGF-β	↑

P: plasma, AH: aqueous humor, VH: vitreous humor. ↑: levels increased in cases when compared to controls; =: no level differences between when comparing cases and controls; ↓: levels decreased in cases when compared to controls.

**Table 3 jcm-13-07426-t003:** Characteristics of prospective studies (*n* = 13).

Reference	Sample Size (*n*)	Age, Ethnicity	Sample Type	Biomarker Type	Biomarker	Outcome
[32]	68	54–68, Indian	AH	Vascular dysfunction	VEGF	↑
				Inflammation	IL-6	↑
					IL-8	↑
[33]	33	40–80, Korean	AH	Vascular dysfunction	VEGF	↑
				Inflammation	IL-6	↑
					IL-8	↑
				microRNA	hsa-miR-15a-5p	↓
					hsa-miR-15b-5p	↑
					hsa-miR-17-5p	↓
					hsa-miR-20a-5p	↓
					hsa-miR-185-5p	↓
				Trophic factor	PlGF	↑
[64]	66	48–64, Turkish	AH	Inflammation	GRP78	↑
[58]	150	40–80, Turkish	P	Inflammation	SII	↑
[44]	49	38–76, Canadian	AH	Vascular dysfunction	ICAM-1	↑
				Inflammation	IL-10	↓
[59]	39	47–69, Japanese	P	Inflammation	Th1/Th2 ratio	↓
[29]	117	48–64 Korean	AH	Vascular dysfunction	VEGF	↑
				Inflammation	GRP78	↑
[55]	79	44–77, Korean	P	Inflammation	sCD14	↑
[75]	392	35–63, Chinese	P	Other	NSE	↑
[34]	88	47–70, Turkish	P	Vascular dysfunction	VEGF	↑
				Oxidative stress	TOS	↑
				Trophic factor	FGF	↑
[50]	159	28–80, Japanese	P	Inflammation	IL-6	↑
[52]	24	45–80, Indian	AH	Inflammation	IL-1RA/IL-1β ratio	↑
[35]	36	47–73, Turkish	AH	Vascular dysfunction	VEGF	↑
				Inflammation	IL-8	↑
				Other	HbA1c	↑

P: plasma, AH: aqueous humor. ↑: levels increased in cases when compared to controls; ↓: levels decreased in cases when compared to controls.

**Table 4 jcm-13-07426-t004:** Characteristics of retrospective studies (*n* = 14).

Reference	Sample Size (*n*)	Age, Ethnicity	Sample Type	Biomarker Type	Biomarker	Outcome
[36]	20	33–87, Greek	VH	Vascular dysfunction	ICAM-1	↑
					VCAM-1	↑
					VEGF	↑
				Inflammation	Lipocalin 2	↑
					IL-6	↑
					IL-8	↑
					IL-27	↑
					TNF-α	↑
[70]	109	30–85, Chinese	P/AH	Apoptosis	Lnc-RNA-SNHG5	↓
[47]	44	43–66, Chinese	P	Vascular dysfunction/microRNA	miR-377-3p	↓
[68]	66	49–82, Turkish	P	Oxidative stress	IMA	↑
					TAS	↓
					TOS	↑
[42]	62	55–75, Indian	AH	Vascular dysfunction	VEGF	=
				Inflammation	IL-6	↑
					IL-8	↑
					IP-10	↑
					MCP-1	=
				Trophic factor	PDGF-AA	↑
[72]	65	42–72, Korean	P/AH	Other	Plasmatic VitD	=
					AH VitD	↑
[62]	31	61–79, Japanese	VH	Inflammation	SAP	↑
					Procalcitonin	↑
					Ferritin	↑
					Fibrinogen	↑
[37]	67	47–66, Korean	AH	Vascular dysfunction	VEGF	↑
				Inflammation	IL-6	↑
					IL-8	↑
				Trophic factor	PlGF	↑
[67]	60	40–67, Korean	P	Inflammation	Oxylipins	↑
				Other	Amino acids	↓
					Citric acid	↓
					Uric acid	↓
[60]	54	52–76, Chinese	AH	Inflammation	ANGPTL4	↑
[76]	83	51–82, Japanese	P	Other	HK1-Ab	↑
[43]	200	47–74, Chinese	P	Inflammation	TIMP-1	↑
					ANG-1	↑
					VEGFR-2	↑
				Trophic factor	PDGF-BB	↑
[53]	81	18–40, Chinese	P	Inflammation	%Neutrophils	↑
					%Monocytes	↓
					%Lymphocytes	↓
[54]	42	46–66, Chinese	P	Inflammation	%Neutrophils	↑
					%Lymphocytes	↓

P: plasma, AH: aqueous humor, VH: vitreous humor. ↑: levels increased in cases when compared to controls; =: no level differences between when comparing cases and controls; ↓: levels decreased in cases when compared to controls.

**Table 5 jcm-13-07426-t005:** Characteristics of cross-sectional studies (*n* = 15).

Reference	Sample Size (*n*)	Age, Ethnicity	Sample Type	Biomarker Type	Biomarker	Outcome
[56]	36	56–72, Greek	P	Inflammation	HWBCC	↑
				Other	Hematocrit	↓
					Lipoprotein	↓
[38]	76	59–78, Spanish	P	Vascular dysfunction	VEGF	↑
				Inflammation	IL-6	↑
					IL-8	↑
					IL-10	↓
[46]	264	18–75, Chinese	P	Vascular dysfunction	ICAM-1	↑
				Inflammation	Hcy	↑
					MCP-1	↑
[39]	76	53–68, Japanese	VH	Vascular dysfunction	ICAM-1	↑
					VEGF	↑
				Inflammation	IL-6	↑
					MCP-1	↑
				Trophic factor	PEDF	↓
[63]	44	51–73, Spanish	VH	Inflammation	EPO	↑
[73]	305	43–68, Chinese	P	Other	Uric acid	↑
[71]	66	30–76, Korean	AH	Other	DKK-3	↑
[40]	80	52–91, German	AH	Vascular dysfunction	VEGF	↑
				Trophic factor	FGF	↑
[48]	394	52–72, Turkish	P	Vascular dysfunction	CHI3L1	↑
				Inflammation	IL-6	↑
[50]	87	54–78, Turkish	P/AH	Inflammation	CRP	↑
					TNF-α	↑
[65]	291	43–68, Chinese	P	Inflammation	Hcy	↑
[74]	100	45–65, Indian	P	Other	Urea	↑
[41]	38	56–76, Japanese	P/AH/VH	Vascular dysfunction	VEGF	↑
				Inflammation	sCD14	↑
					IL-8	↑
					IP-10	↑
					MCP-1	↑
					MIG	↑
[61]	172	49–72, Chinese	P	Inflammation	ANGPTL4	↑
[45]	45	64–84, Chinese	AH	Vascular dysfunction	MCP-1	↑
				Inflammation	ICAM-1	↑

P: plasma, AH: aqueous humor, VH: vitreous humor. ↑: levels increased in cases when compared to controls; ↓: levels decreased in cases when compared to controls.

## 4. Discussion

DME is a prevalent complication in DM patients. From a pathophysiological perspective, DME is characterized by a complex interplay of various mechanisms that involve a variety of molecular-genetic actors. This systematic review underscores the significance of these biological particles and their utility as biomarkers to diagnose and manage DME.

The predominant role of inflammation in DME is evident from the reviewed studies. Cytokines/Chemokines and growth factors, such as IL-1α2 [31], IL-6 [30,31,32,33,36,37,38,39,42,48,49,50], IL-8 [31,32,33,35,36,37,38,41,42], IL-27 [36], TNF-α [36,51], IP-10 [31,41,42], MCP-1 [31,39,41,42,45,46], and MIG [31,41], were frequently reported with increased levels in various biological fluids, suggesting their involvement in the inflammatory cascade associated with DME. The decreased levels of IL-10 [38,44] highlight the complexity of cytokine interactions, potentially indicating a lack of anti-inflammatory response in DME patients. Additionally, an elevated IL-1RA/IL-1β ratio in vitreous humor further supports the pro-inflammatory conditions that characterize the pathology [52]. These results are in line with the described inflammatory nature of DM [77] and, specifically, DR [78]. In this sense, the presence of similar inflammatory biomarkers in patients with these conditions proves that both visual pathologies share a strong pathogenic origin, making these biomarkers less specific.

The importance of the inflammatory response is also reflected by the cellular biomarkers extracted from the reviewed studies, including neutrophil, monocyte and lymphocyte counts [53,54]. Furthermore, an increased CD14 expression was linked to DME, indicating the presence of systemic inflammation and macrophage activation [41,54]. The increased white blood cell count [56], mean platelet volume, neutrophil-to-lymphocyte ratio, and systemic immune inflammation index also emphasize the systemic inflammatory response [57,58], as similarly reported for DM [79,80] and DR [81,82]. Moreover, the discovery of a reduced Th1/Th2 ratio highlights the importance of a skewed immune response towards inflammation [59], as similarly described in diabetic patients and subjects with DR [83].

The finding of increased levels of soluble inflammation-related molecules, such as ANG-1 [43], TIMP-1, ANGPLT4 [60,61], ferritin, fibrinogen, procalcitonin, SAP [62], MMPs, PAI-1 [31], CRP [49,51], EPO [63], GRP78 [29,64], Hcy [46,65], and lipocalin 2 [36], suggests their potential roles in the development and progression of DME. In relation to this, there were no differences in the plasmatic levels of ANG-1 between diabetic and healthy individuals [84], making this molecule a potential and specific biomarker for DME. Similarly, significantly lower ANG-1 serum levels were reported when comparing the DR group to the non-DR one [85], which contrasts the results included in this review regarding DME. Also, levels of serum MMP-2 and CRP were raised in patients with DR when compared to those without the pathology, whereas the serum TIMP-1 level was reduced [86], as opposed to the results obtained by the studies included in this systematic review. However, TIMP-1, MMP-3, and MMP-9 levels were found to be increased in the aqueous humor samples of DR patients when compared to non-diabetic ones [87], implying the possibility that the measurement of these molecules in eye-derived samples could be specific for certain visual conditions. Other studies have shown that GRP78 [88], Hcy [89,90] and lipocalin 2 [91,92], are increased in several biological fluids of DR patients, agreeing with the findings related to DME collected in this systematic review.

The altered levels of lipid-related pro-inflammatory molecules, including oxylipins [66] and various lipid metabolites [67], further highlight the importance of metabolic disturbances contributing to the inflammatory process in DME. These results are similar to the ones found in DR patients [93,94], showing the potential utility of these molecules as biomarkers to control the onset and progression of these pathologies.

OS markers were identified in a subset of studies, indicating their role in DME pathogenesis. Elevated levels of MDA, 4-HNE, SOD [30], IMA [68], and TOS [34,68], along with decreased CAT and TAS levels [30,68], suggest an imbalance between oxidative and antioxidative mechanisms. This prooxidant milieu likely contributes to retinal damage and the progression of DME, being in conformity with the findings obtained in studies with DR patients [95,96,97].

Another group of biomarkers included in the systematic review were the miRNAs, which are emerging as crucial regulators in various diseases, including DM complications [98]. In fact, it has been shown that OS mediates epigenetic modifications, via specific miRNAs, and the expression of genes related to apoptosis in DR patients [99]. The reviewed studies showed decreased levels of specific miRNAs, such as let-7c-5p, hsa-miR-200b-3p, hsa-miR-199a-3p, hsa-miR-365-3 [69], hsa-miR-377-3p [47], hsa-miR-185-5p, hsa-miR-17-5p, hsa-miR-20a-5p, hsa-miR-15b-5p, and hsa-miR-15a-5p [33], suggesting their potential role in the pathophysiology of DME. Dysregulation in the levels of some of these miRNAs has been described in several studies. For instance, downregulation of has-miR-199a-3p has been involved in the progression of endometriosis, affecting the function of endothelial cells in a similar way in which could happen in DME [100]. Bartoszewski et al. conducted a study in which they proved that miR-200b has a significant role in the angiogenic phenomenon [101], which could corroborate its participation in DME. Also, Yuan et al. have shown that hsa-miR-377-3p is involved in cellular proliferation processes [102], being a potential actor in angiogenesis. Finally, a study conducted in human primary retinal endothelial cells showed that hsa-miR-15a-5p prevents the permeability capillary dysfunction of retinal vasculature [103], explaining why a decrease of this miRNA could be found in DME patients and could be considered as a specific biomarker of the pathology. All the mentioned miRNAs could serve as non-invasive biomarkers for the early detection and monitoring of DME, although more studies with bigger sample sizes are still required to validate them as a diagnostic and prognostic tool.

Vascular dysfunction is also a hallmark of diabetic complications, including DME. The significant increase in VEGF levels across various studies underscores its pivotal role in vascular permeability and angiogenesis in DME [29,30,31,32,33,34,35,36,37,38,39,40,41], being a common hallmark of other diabetic complications such as DR [104] and diabetic nephropathy [105]. The elevated levels of VEGFR2 [43], ICAM-1 [31,36,39,44,45,46], VCAM-1 [31,36], and other less common vascular-related molecules further support the involvement of endothelial dysfunction and vascular activation in DME.

The apoptotic process in DME was highlighted by increased levels of CAS-3 [30] and decreased levels of lncRNA SNHG5 [70]. CAS-3 is a principal actor on the apoptotic process [106], whereas lncRNA SNHG5 is a molecule that has been shown to regulate cell viability and apoptosis in several cell types [107,108,109]. These findings suggest that apoptosis plays an important role in the pathophysiology of DME, such as it happens in DM [110] and DR [111,112].

Trophic factors, such as PlGF [31,33,37], FGF [34,40], EGF, HGF, TGF-β, and PDGF [42,43], were associated with DME, indicating a potential role in the pathology. The maintenance of an equilibrium on the level of these molecules is crucial to maintaining the correct homeostasis of the retinal tissue [113] and its monitoring could identify potential biomarkers for disease severity and progression.

This work also identified several biomarkers not classified into the previous groups, such as DKK-3 [71], CA [66], various amino acids [66,67], vitamin D [72], uric acid [67], citric acid [67,73], urea [74], HbA1c [35], NSE [75], and HK1-Ab [76]. These molecules may constitute a direct reflection of the metabolic disturbances and diverse pathological processes involved in DME, providing a broader understanding of the disease. Similarly, other works have highlighted the role of some of these molecular agents in other related pathologies, especially DR [114,115,116]. However, additional data is needed to confirm the potential use of these metabolites as biomarkers for the diagnosis and management of retinal vascular diseases.

The presented review has several notable strengths. One of the primary strengths is the inclusion of studies conducted on diverse populations. This diversity enhances the external validity of the findings, making them more applicable to a broader range of individuals and increasing the generalizability of the results. Additionally, the review specifically includes studies focused on the pathology in question, while excluding experimental animal studies. This focus ensures that the findings are directly relevant to human disease and clinical practice. Moreover, the inclusion of studies investigating different molecular types allows for a more comprehensive understanding of the disease’s pathophysiological aspects. By examining various molecular markers, the review provides a richer and more nuanced insight into the disease mechanisms, which could inform future research and clinical strategies. Despite its strengths, this work encounters several limitations that should be acknowledged. Firstly, some of the included studies focus exclusively on a single sample type, which may limit the generalizability of the findings across different biological matrices. Secondly, the sample size in certain studies is insufficient to draw robust conclusions about the validity and reliability of the biomarkers under investigation. This small sample size issue hampers the statistical power and could lead to biased outcomes. Finally, there is significant heterogeneity in study designs, including variations in methodologies, participant selection criteria, and analytical techniques. This variability makes it challenging to compare results across studies directly and may affect the overall synthesis and interpretation of the findings.

As a summary, Table 6 underscores a collection of the most frequently mentioned biomarkers included in the reviewed studies. We propose VEGF, IL-6, IL-8, MCP-1, ICAM-1, and PlGF as potential tools to improve DME’s follow up and management. Remarkably, most of these molecules are involved in the inflammatory event that characterizes DME, thoroughly described in this systematic review. This highlights the importance of this phenomenon in the disease progression and opens a window for the potential identification of new therapeutic targets.

## 5. Conclusions

This work underscores the complex and multifactorial nature of DME, involving inflammation, OS, vascular dysfunction, apoptosis, and the imbalance of trophic factors. These biomarkers in general, and more concretely VEGF, IL-6, IL-8, and MCP-1, could offer valuable insights into the underlying mechanisms of DME and hold promises for improving diagnostic accuracy, disease monitoring, and developing targeted therapies. Further research is warranted to validate these biomarkers and explore their potential clinical applications in managing DME.

## Figures and Tables

**Figure 1 jcm-13-07426-f001:**
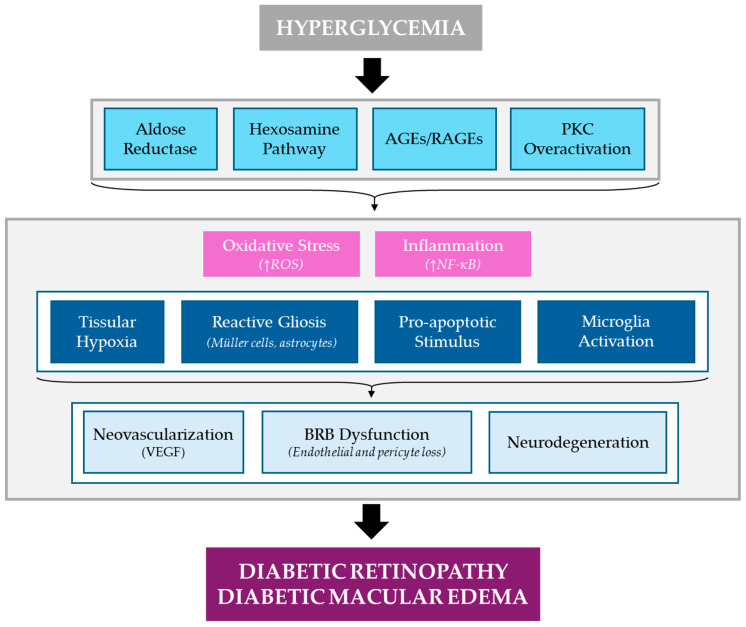
Molecular pathways involved in DR/DME onset and progression. Hyperglycemic conditions found in DM patients lead to the activation of several glucose-related biochemical pathways, involving the activity of aldose reductase and hexosamine, the overactivation of protein kinase C (PKC), and the formation of advanced glycation end products (AGEs) which exert their effects by binding to specific receptors (RAGEs). The activity of these molecular processes results in the establishment of a pro-oxidant and pro-inflammatory response characterized, amongst other facts, by the increase of reactive oxygen species (ROS) production and the translocation of nuclear factor kappa B (NF-κB) into the retina cells nuclei, with the consequent synthesis of pro-inflammatory mediators. These phenomena lead to a series of changes in the physiology of the retina (tissular hypoxia, reactive gliosis, pro-apoptotic stimulus, and microglia activation) which, in turn, induce the clinical signs of DR/DME: (i) neovascularization as a consequence of an over-expression of vascular endothelial growth factor (VEGF), (ii) blood-retinal barrier (BRB) dysfunction due to a loss of endothelial cells and pericytes, and (iii) degeneration of neural cells. DR: diabetic retinopathy, DME: diabetic macular edema, DM: diabetes mellitus.

**Figure 2 jcm-13-07426-f002:**
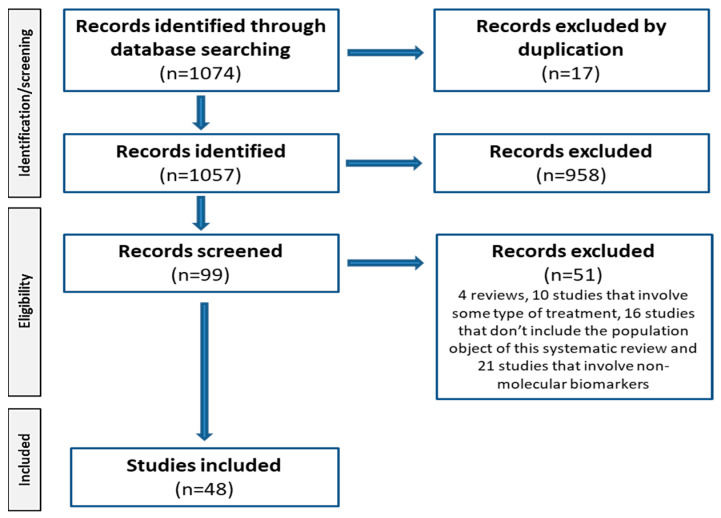
PRISMA flow diagram for the systematic review detailing the records identified through database searching, the records excluded in the different phases of the selection process, and the full texts included in the systematic review.

**Figure 3 jcm-13-07426-f003:**
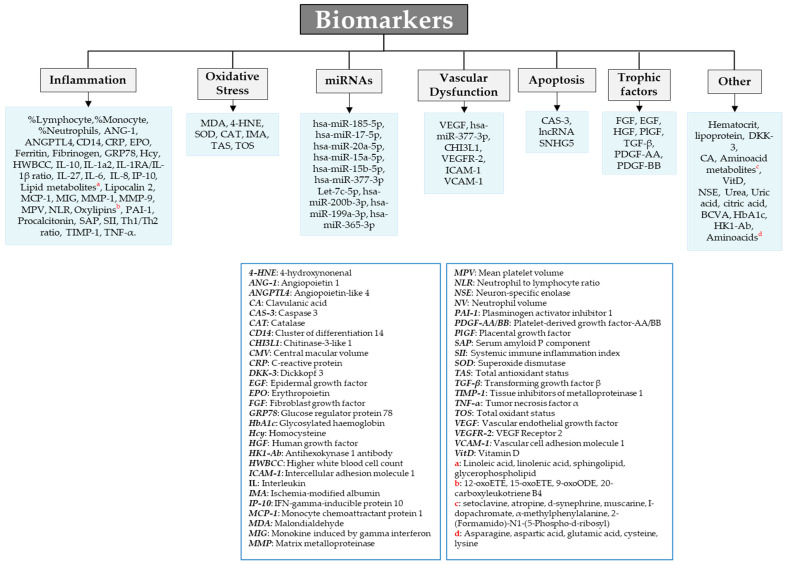
Main cellular and molecular-genetic biomarkers. An overview based on the studies included in the systematic review.

**Table 1 jcm-13-07426-t001:** Global evolution of diabetic population.

Country/Territory	2000	2011	2021	2030	2045
Africa	2.5	14.7	23.6	33.4	55.3
Europe	22.4	52.8	61.4	67.0	69.0
Middle East and North Africa	17.0	32.6	72.7	95.0	135.7
North America and Caribbean	21.4	37.7	50.5	57.0	63.0
South and Central America	8.6	25.1	32.5	40.0	49.0
South-East Asia	34.9	71.4	90.2	113.3	151.5
Western Pacific	44.1	131.9	205.6	238.3	260.2
Total	150.9	366.2	536.5	644.0	783.7

Data refers to millions of people.

**Table 6 jcm-13-07426-t006:** Main DME-related molecular biomarkers identified in the systematic review.

Molecule	Number of Studies	Sample Type	References
VEGF	14	AH	[29,32,33,35,37,40,41,42]
		VH	[36,39,41]
		P	[30,34,38,41]
IL-6	12	AH	[31,32,33,37,42]
		VH	[30,36,39]
		P	[30,38,48,49,50]
IL-8	9	AH	[31,32,33,35,37,41,42]
		VH	[36,41]
		P	[38,41]
MCP-1	8	AH	[31,41,42,45]
		VH	[39,41]
		P	[41,46]
ICAM-1	6	AH	[31,44,45]
		VH	[36,39]
		P	[46]
PlGF	3	AH	[31,33,37]

P: plasma, AH: aqueous humor, VH: vitreous humor.

## Data Availability

Data extraction process can be checked in the Section 2.

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
