# Peer review of "Molecular-Genetic Biomarkers of Diabetic Macular Edema"

_jcm, 2024, doi:10.3390/jcm13237426_

Round 1

Reviewer 1 Report

Comments and Suggestions for Authors

In recent years, a great effort has been made to understand the molecular mechanisms in the pathogenesis of DME; this systematic review, updated to 2023, presents the main biomarkers found in DME and offers an updated overview of the research.

There are some comments about manuscrit:

In the Introduction (from line 85 to 94) the description of the manifestation of DME and DR is a bit confusing.

Furthermore, from line 96, a differentation shoud be made between screening tests  and other more invasive examinations such as fluorescein angiography

From line 105 it should be pointed that a variety of treatment are currently available for DME patients, otherwise it is not clear. At the end of the sentence reference 4 is not correct.

In the last sentence it should be emphasized that biomarkers refer to plasma, aqueus humor or vitreous samples.

Inclusion-exclusion criteria: Authors affirm they excluded studies with DME patients younger than 40 years old, but the age reported for some studies in Tables 2-5 is less than 40 (and some studies do not specify in methods if patiens were affected by type 1 or 2 diabetes)

Quality assessment: Authors assessed the studies included in their review and they should show their results ( in Supplements o in a new figure)

Main biomarker in DME (line 237): to say that "...the importance of different serological, cellular-related and molecular biomarkers for the diagnosis and management ..." is too assertive: we still do not have the tools to establish DME diagnosis and therapy through molecular markers. 

Line 246: "and one study reporting no difference in the aqueous humor expression of VEGF". In the study of Kim et al 2015 no difference in VEGF levels was foud among different type of DME patients, but DME patients showed higher amounts of VEGF versus controls.

Regarding limitations, it should be underlined that in the various studies the control cases were different, being alternatively represented by non-diabetic patients, by patients with type 2 diabetes without DR or with DR and no DME.

Author Response

ANSWERS TO THE REVIEWERS

The authors of this Review thank The Reviewers and The Editor of the Journal of Clinical Medicine, for outstanding review work. We also want to thank The Reviewers and The Editor for the opportunity to clarify our research, as well as to improve our manuscript. Every concern The Reviewers have raised, has been addressed by the authors. The present revised version of our work is now sent to the Journal of Clinical Medicine, for evaluation.

We have responded point by point to the comments and suggestions. We have also reviewed the English language through the ms., and illustrations, as suggested.

The corresponding answers of the authors to The Reviewers are listed below.

REVIEWER 1

Open Review

Comments and Suggestions for Authors

In this manuscript, many molecular biomarkers of DME are listed. This data summary is important for the further research regarding drug development in DME.

But one major issue is related to "what is the key marker to affect DME progress?".

The authors thank The Reviewer for calling out attention to this important issue. Our response is given below:

There are different key biomarkers that play an important role in DME development, depending on which signaling pathway is being studied (inflammation, oxidative stress, angiogenesis, etc.). However, our research work determined that VEGF, IL-6, IL-8, MCP-1, ICAM-1 and PlGF are the ones that could be considered pivotal when studying DME. However, an additional paragraph has been added to the discussion section (lines 460-466) addressing this idea, along with the Table 6, which collects the number of studies that mention these molecules and the sample type used. We believe that, as pointed out by the reviewer, this addition strengthens the understanding of the manuscript and directly answers the question of which markers seem to be more important.

Although this summary can understand what kinds of markers should be checked or considered for DME progression, mentioning the major issue above and discussing it in detail should be essential.

            The authors thank The Reviewer for raising this topic.

As mentioned above, the new text addition to the discussion section and Table 6 should address this question.

Is it possible to analyze each human study's age/sample type/sample size/biomarker in all-in-one to see what is the most importantly or frequently mentioned molecule or the list of them with the important order?

We fully agree with The Reviewer in this concern. According to these comments, we have been changed some sentences, and the results can be checked in Table 6.

One minor point is to add gender and ethnicity in the table.

According to The Reviewer suggestion, and, as mentioned in the inclusion and exclusion criteria (lines 148-165), the studies considered for this systematic review included patients from both sexes, and every study included patients from both sexes. In agreement with this idea, the information related to the ethnicity of the study participants has been added to the results tables.  

Furthermore, it should be accessible to the original paper in table adding references in each study. This is because providing name and year only is reader unfriendly. 

The authors are extremely grateful to The Reviewer for this important suggestion. This has been changed and now tables have the reference number instead of the first author’s name and year.

Reviewer 2 Report

Comments and Suggestions for Authors

In this manuscript, many molecular biomarkers of DME are listed. This data summary is important for the further research regarding drug development in DME. But, one major issue is related to "what is the key marker to affect DME progress?". Although this summary can understand what kinds of markers should be checked or considered for DME progression, mentioning the major issue above and discussing it in detail should be essential.

Is it possible to analyze each human study's age/sample type/sample size/biomarker in all-in-one to see what is the most importantly or frequently mentioned molecule or the list of them with the important order?

One minor point is to add gender and ethnicity in the table.

Furthermore, it should be accessible to the original paper in table adding references in each study. This is because providing name and year only is reader-unfriendly. 

Author Response

ANSWERS TO THE REVIEWERS

The authors of this Review thank The Reviewers and The Editor of the Journal of Clinical Medicine, for outstanding review work. We also want to thank The Reviewers and The Editor for the opportunity to clarify our research, as well as to improve our manuscript. Every concern The Reviewers have raised, has been addressed by the authors. The present revised version of our work is now sent to the Journal of Clinical Medicine, for evaluation.

We have responded point by point to the comments and suggestions. We have also reviewed the English language through the ms., and illustrations, as suggested.

The corresponding answers of the authors to The Reviewers are listed below.

REVIEWER 2

Open Review

Comments and Suggestions for Authors

In the study “Molecular Biomarkers of Diabetic Macular Edema” by Andres-Blasco et al., the authors present a systematic review in order to summarize current knowledge on molecular biomarkers associated with DM, with the focus on their diagnostic potential. They state that they have identified various molecules that exhibit differential expression profiles in different biological fluids when comparing diabetic individuals with or without macular edema.

This study is thorough and methodology that is applied is described in detail. It merits publication but some issues require further explanation.

The authors meticulously described findings that they have presented in Tables 2-5 in the Result section of the MS. However, in the Discussion part it is expected to draw some conclusions about the presented data. It is very hard for the reader to conclude which are the specific biomarkers (if any) for DM and if these biomarkers are also common (or not) for DR. Some of the changes were compared between the diabetic patients and the healthy controls. The other changes are evaluated between the diabetic patients with or without the DR.

We appreciate the feedback and the advice of The Reviewer. An additional text has been added to address this, including the most mentioned markers extracted from the included studies (lines 460-466, Table 6). Most of these markers are involved in the inflammatory process that characterizes DME.

The title is, thus, somewhat misleading. It implies that biomarkers for DM have been identified. If so, a Table summarizing the data in terms of specificity for DM will be of great help for the reader. If this is not the case, and some changes have the potential to be considered as biomarkers that should be reflected in the title as well.

As kindly suggested by The Reviewer, we have slightly modified the title to: “Molecular-Genetic Biomarkers of Diabetic Macular Edema”. Also, according to the suggestions of The Reviewer, we have improved the Table 6 by adding the most mentioned biomarkers in the included studies in this work (lines 467-468). Thank you so much for outstanding help.

FINAL COMMENTS

We agree with The Reviewers of this work in all points raised, as well as for the carved time out of their busy schedules and evaluating our work.

We have learned from every comment and suggestion from The Reviewers. The English Language has been carefully reviewed by an expert. Consequently, we tried to do a more precise sentences through the text and illustrations.

Thank you so much to The Reviewers and The Editor for offering constructive criticism and detailing our article limitations.

Reviewer 3 Report

Comments and Suggestions for Authors

In the study ”Molecular Biomarkers of Diabetic Macular Edema” by Andres-Blasco et al., the authors present a systematic review in order to summarize current knowledge on molecular biomarkers associated with DM, with the focus on their diagnostic potential. They state that they have identified various molecules that exhibit differential expression profiles in different biological fluids when comparing diabetic individuals with or without macular edema.

This study is thorough and methodology that is applied is described in detail. It merits publication but some issues require further explanation.

The authors meticulously described findings that they have presented in Tables 2-5 in the Result section of the MS. However, in the Discussion part it is expected to draw some conclusions about the presented data. It is very hard for the reader to conclude which are the specific biomarkers (if any) for DM and if these biomarkers are also common (or not) for DR. Some of the changes were compared between the diabetic patients and the healthy controls. The other changes are evaluated between the diabetic patients with or without the DR.

The title is, thus, somewhat misleading. It implies that biomarkers for DM have been identified. If so, a Table summarizing the data in terms of specificity for DM will be of great help for the reader. If this is not the case, and some changes have the potential to be considered as biomarkers that should be reflected in the title as well.

Author Response

ANSWERS TO THE REVIEWERS

The authors of this Review thank The Reviewers and The Editor of the Journal of Clinical Medicine, for outstanding review work. We also want to thank The Reviewers and The Editor for the opportunity to clarify our research, as well as to improve our manuscript. Every concern The Reviewers have raised, has been addressed by the authors. The present revised version of our work is now sent to the Journal of Clinical Medicine, for evaluation.

We have responded point by point to the comments and suggestions. We have also reviewed the English language through the ms., and illustrations, as suggested.

The corresponding answers of the authors to The Reviewers are listed below.

REVIEWER 1

Open Review

Comments and Suggestions for Authors

In this manuscript, many molecular biomarkers of DME are listed. This data summary is important for the further research regarding drug development in DME.

But one major issue is related to "what is the key marker to affect DME progress?".

The authors thank The Reviewer for calling out attention to this important issue. Our response is given below:

There are different key biomarkers that play an important role in DME development, depending on which signaling pathway is being studied (inflammation, oxidative stress, angiogenesis, etc.). However, our research work determined that VEGF, IL-6, IL-8, MCP-1, ICAM-1 and PlGF are the ones that could be considered pivotal when studying DME. However, an additional paragraph has been added to the discussion section (lines 460-466) addressing this idea, along with the Table 6, which collects the number of studies that mention these molecules and the sample type used. We believe that, as pointed out by the reviewer, this addition strengthens the understanding of the manuscript and directly answers the question of which markers seem to be more important.

Although this summary can understand what kinds of markers should be checked or considered for DME progression, mentioning the major issue above and discussing it in detail should be essential.

            The authors thank The Reviewer for raising this topic.

As mentioned above, the new text addition to the discussion section and Table 6 should address this question.

Is it possible to analyze each human study's age/sample type/sample size/biomarker in all-in-one to see what is the most importantly or frequently mentioned molecule or the list of them with the important order?

We fully agree with The Reviewer in this concern. According to these comments, we have been changed some sentences, and the results can be checked in Table 6.

One minor point is to add gender and ethnicity in the table.

According to The Reviewer suggestion, and, as mentioned in the inclusion and exclusion criteria (lines 148-165), the studies considered for this systematic review included patients from both sexes, and every study included patients from both sexes. In agreement with this idea, the information related to the ethnicity of the study participants has been added to the results tables.  

Furthermore, it should be accessible to the original paper in table adding references in each study. This is because providing name and year only is reader unfriendly. 

The authors are extremely grateful to The Reviewer for this important suggestion. This has been changed and now tables have the reference number instead of the first author’s name and year.

REVIEWER 2

Open Review

Comments and Suggestions for Authors

In the study “Molecular Biomarkers of Diabetic Macular Edema” by Andres-Blasco et al., the authors present a systematic review in order to summarize current knowledge on molecular biomarkers associated with DM, with the focus on their diagnostic potential. They state that they have identified various molecules that exhibit differential expression profiles in different biological fluids when comparing diabetic individuals with or without macular edema.

This study is thorough and methodology that is applied is described in detail. It merits publication but some issues require further explanation.

The authors meticulously described findings that they have presented in Tables 2-5 in the Result section of the MS. However, in the Discussion part it is expected to draw some conclusions about the presented data. It is very hard for the reader to conclude which are the specific biomarkers (if any) for DM and if these biomarkers are also common (or not) for DR. Some of the changes were compared between the diabetic patients and the healthy controls. The other changes are evaluated between the diabetic patients with or without the DR.

We appreciate the feedback and the advice of The Reviewer. An additional text has been added to address this, including the most mentioned markers extracted from the included studies (lines 460-466, Table 6). Most of these markers are involved in the inflammatory process that characterizes DME.

The title is, thus, somewhat misleading. It implies that biomarkers for DM have been identified. If so, a Table summarizing the data in terms of specificity for DM will be of great help for the reader. If this is not the case, and some changes have the potential to be considered as biomarkers that should be reflected in the title as well.

As kindly suggested by The Reviewer, we have slightly modified the title to: “Molecular-Genetic Biomarkers of Diabetic Macular Edema”. Also, according to the suggestions of The Reviewer, we have improved the Table 6 by adding the most mentioned biomarkers in the included studies in this work (lines 467-468). Thank you so much for outstanding help.

FINAL COMMENTS

We agree with The Reviewers of this work in all points raised, as well as for the carved time out of their busy schedules and evaluating our work.

We have learned from every comment and suggestion from The Reviewers. The English Language has been carefully reviewed by an expert. Consequently, we tried to do a more precise sentences through the text and illustrations.

Thank you so much to The Reviewers and The Editor for offering constructive criticism and detailing our article limitations.

Round 2

Reviewer 3 Report

Comments and Suggestions for Authors

I think that the authors have adequately addressed the comments made by the reviewers in the revised version of the manuscript. Therefore, I have no further comments.

Author Response

The following changes have been enclosed according to your kind recommendation.   1. The fully completed PRISMA checklist for your systematic review.
  On Page 4 (Methods) we can enclosed a new paragraph   Using the PRISMA checklist (Figure 2), we conducted a systematic review of all studies published in peer-reviewed journals. We initially retrieved 1074 articles written in English from Medline (via PubMed), Embase, and Cochrane Library with a date range from 2004 to 2023. We systematically searched the database by combining the following keywords: biomarker AND [caspase OR apoptosis OR oxidative stress OR neurodegeneration OR cytokine OR VEGF OR inflammation OR vascular dysfunction OR microRNA OR miRNA OR neurotrophic factor OR growth factor] AND [diabetic macular edema OR diabetic macular oedema]. 17 of the total initial publications were excluded due to duplication reasons. From the 1057 identified studies, 958 were excluded for not following the inclusion criteria cited in subsection 2.3. The second phase of eligibility resulted on the exclusion of 51 of the 99 pre-selected studies since they involved some type of treatment, did not include the population object of the systematic review or involved exclusively non-molecular biomarkers. The following of the PRISMA checklist resulted in the final selection of 48 studies that were included in the systematic review.   2. A statement in the Methods section confirming adherence to the PRISMA guidelines, and if applicable, the registration number and date of your systematic review.
  This has been done by incorporating a new sentence   We have reviewed again our text and illustrations and the final version of our work is enclosed below
